# Endoplasmic Reticulum Stress Promotes the Expression of TNF-α in THP-1 Cells by Mechanisms Involving ROS/CHOP/HIF-1α and MAPK/NF-κB Pathways

**DOI:** 10.3390/ijms242015186

**Published:** 2023-10-14

**Authors:** Nadeem Akhter, Ajit Wilson, Hossein Arefanian, Reeby Thomas, Shihab Kochumon, Fatema Al-Rashed, Mohamed Abu-Farha, Ashraf Al-Madhoun, Fahd Al-Mulla, Rasheed Ahmad, Sardar Sindhu

**Affiliations:** 1Department of Immunology & Microbiology, Dasman Diabetes Institute, P.O. Box 1180, Dasman 15462, Kuwait; nadeem.akhter@dasmaninstitute.org (N.A.); ajit.wilson@dasmaninstitute.org (A.W.); hossein.arefanian@dasmaninstitute.org (H.A.); reeby.thomas@dasmaninstitute.org (R.T.); shihab.kochumon@dasmaninstitute.org (S.K.); fatema.alrashed@dasmaninstitute.org (F.A.-R.); rasheed.ahmad@dasmaninstitute.org (R.A.); 2Department of Translational Research, Dasman Diabetes Institute, P.O. Box 1180, Dasman 15462, Kuwait; mohamed.abufarha@dasmaninstitute.org (M.A.-F.); fahd.almulla@dasmaninstitute.org (F.A.-M.); 3Department of Genetics & Bioinformatics, Dasman Diabetes Institute, P.O. Box 1180, Dasman 15462, Kuwait; ashraf.madhoun@dasmaninstitute.org; 4Animal & Imaging Core Facilities, Dasman Diabetes Institute, P.O. Box 1180, Dasman 15462, Kuwait

**Keywords:** ER stress, metabolic stress, obesity, metabolic syndrome, inflammation, TNF-α, ROS, CHOP, HIF-1α, MAPK/NF-κB

## Abstract

Obesity and metabolic syndrome involve chronic low-grade inflammation called metabolic inflammation as well as metabolic derangements from increased endotoxin and free fatty acids. It is debated whether the endoplasmic reticulum (ER) stress in monocytic cells can contribute to amplify metabolic inflammation; if so, by which mechanism(s). To test this, metabolic stress was induced in THP-1 cells and primary human monocytes by treatments with lipopolysaccharide (LPS), palmitic acid (PA), or oleic acid (OA), in the presence or absence of the ER stressor thapsigargin (TG). Gene expression of tumor necrosis factor (*TNF*)-*α* and markers of ER/oxidative stress were determined by qRT-PCR, TNF-α protein by ELISA, reactive oxygen species (ROS) by DCFH-DA assay, hypoxia-inducible factor 1-alpha (HIF-1α), p38, extracellular signal-regulated kinase (ERK)-1,2, and nuclear factor kappa B (NF-κB) phosphorylation by immunoblotting, and insulin sensitivity by glucose-uptake assay. Regarding clinical analyses, adipose TNF-α was assessed using qRT-PCR/IHC and plasma TNF-α, high-sensitivity C-reactive protein (hs-CRP), malondialdehyde (MDA), and oxidized low-density lipoprotein (OX-LDL) via ELISA. We found that the cooperative interaction between metabolic and ER stresses promoted TNF-α, ROS, CCAAT-enhancer-binding protein homologous protein (*CHOP*), activating transcription factor 6 (*ATF6*), superoxide dismutase 2 (*SOD2*), and nuclear factor erythroid 2-related factor 2 (*NRF2*) expression (*p* ≤ 0.0183),. However, glucose uptake was not impaired. TNF-α amplification was dependent on HIF-1α stabilization and p38 MAPK/p65 NF-κB phosphorylation, while the MAPK/NF-κB pathway inhibitors and antioxidants/ROS scavengers such as curcumin, allopurinol, and apocynin attenuated the TNF-α production (*p* ≤ 0.05). Individuals with obesity displayed increased adipose TNF-α gene/protein expression as well as elevated plasma levels of TNF-α, CRP, MDA, and OX-LDL (*p* ≤ 0.05). Our findings support a metabolic–ER stress cooperativity model, favoring inflammation by triggering TNF-α production via the ROS/CHOP/HIF-1α and MAPK/NF-κB dependent mechanisms. This study also highlights the therapeutic potential of antioxidants in inflammatory conditions involving metabolic/ER stresses.

## 1. Introduction

Tumor necrosis factor-α (TNF-α) is a proinflammatory cytokine expressed through monocytes and macrophages in response to challenges from endotoxins or bacterial lipopolysaccharides (LPS). TNF-α plays a key role in host immunity as well as immunosurveillance. TNF-α overexpression and increased circulatory levels may lead to detrimental effects associated with both acute and chronic inflammatory conditions and initiation or exacerbation of several types of malignancies. As one of the most important pro-inflammatory cytokines, TNF-α plays roles in vasodilatation, edema formation, and adhesion of leukocytes to the epithelium through expression of adhesion molecules; it regulates blood coagulation, as well as contributing to oxidative stress at sites of inflammation [1,2]. Its overproduction plays significant roles in several diseases of the respiratory [3], cardiovascular [4], nervous [5], skeletal [6], endocrine [7], and metabolic systems [8].

The characteristic chronic low-grade inflammation observed in obesity and metabolic syndrome is called metabolic inflammation and is regarded as a key player in adipose dysfunction and metabolic syndrome [9]. As a prototypic inflammatory cytokine, TNF-*α* plays critical roles in various components of metabolic syndrome and substantial evidence links TNF-*α* with the impaired glucose tolerance and insulin resistance in individuals with obesity [10] and type 2 diabetes (T2D) [11,12], as well as in animal models of these two metabolic disorders [13].

Oxidative stress, which is defined as an imbalance between free radical production and antioxidant defenses, is emerging as a common pathway that links obesity with related complications. Free radicals are reactive pro-oxidant agents to proteins, carbohydrates, and lipids. Antioxidant defenses are pivotal to maintain homeostasis and can be enzymatic viz. superoxide dismutase (SOD), catalase (CAT), glutathione peroxidase (GPx), glutathione S-transferase (GST), thioredoxin (TRX), and peroxiredoxin (PRX) and non-enzymatic (glutathione, vitamins A, C, and E, carotenoids, polyphenols, transferrin, and lipoic acid) [14]. It is speculated that oxidative stress can be both a consequence and a trigger of obesity. On one hand, excessive intake of carbohydrates and fats, especially saturated fatty acids, leads to increased oxidative stress [15]; meanwhile, on the other hand, oxidative stress may promote obesity via favoring pre-adipocyte/adipocyte growth and differentiation [16].

The mechanisms of induction of metabolic inflammation in the expanding adipose tissue in obesity are debated and suggested as including dysregulation in free fatty acid (FFA) fluxes, activation of toll-like receptor (TLR)-4 by fatty acids, adipose tissue hypoxia, oxidative and endoplasmic reticulum (ER) stresses, inhibition of adipogenesis, lipolysis, fibrosis, increased free radical activities, as well as activation of monocytes and macrophages via adipocyte death [17,18]. Given the evidence that human and murine adipocytes after hypoxia induction secrete several adipocytokines, including leptin, visfatin, vascular endothelial growth factor (VEGF), and IL-6 [19,20], as well as the observations that overnutrition or a positive energy balance favors the ER stress [21,22,23], it is likely that ER stress could act as a co-player with lipotoxic stress to amplify metabolic inflammation.

Circulatory levels of LPS and saturated/unsaturated free fatty acids, such as palmitic acid (PA) and oleic acid (OA), are found to be upregulated in obesity and metabolic syndrome [24,25]; meanwhile, these changes were also found to correlate with chronic low-grade inflammation, endotoxemia, hyperlipidemia, and insulin resistance [26,27]. It is known that TNF-α expression in obesity plays a key role in induction of chronic low-grade inflammation and insulin resistance [10,13,28]. However, it remains elusive whether the metabolic (such as exposure to LPS, PA, or OA) and ER stresses, as found under conditions of obesity or metabolic syndrome, can cooperate to amplify the expression of TNF-α in monocytic cells. We hypothesized that such cooperation existed as an underlying driver of TNF-α expression in monocytic cells. Therefore, in this study, we investigated this cooperation mechanism between the ER and metabolic stresses that might play a role in supporting inflammation in metabolic disease conditions. Herein, we show that the ER stress induced by thapsigargin (TG; a non-competitive inhibitor of sarco-ER Ca^2+^ ATPase or SERCA) has an additive effect with metabolic stress induced by LPS, PA, or OA, resulting in increased expression of TNF-α in THP-1 cells via the ROS/CHOP/HIF-1α and MAPK/NF-κB dependent mechanisms.

## 2. Results

### 2.1. The Cooperation between ER and Metabolic Stresses Promotes the Expression of TNF-α Transcripts and Protein in THP-1 Cells

Adipose tissue expansion in obesity is marked by chronic low-grade inflammation from inflammatory cytokines and free fatty acids, as well as oxidative stress from tissue hypoxia. However, it is unclear whether the ER stress could promote TNF-α expression in monocytic cells that are challenged by lipotoxic or metabolic stress. To this end, our data show the upregulated *TNF-α* mRNA expression in THP-1 cells following stimulations with LPS+TG (15.16 ± 0.47 fold), PA+TG (8.40 ± 0.45 fold), and OA+TG (8.0 ± 0.21 fold), compared, respectively, with LPS (10.93 ± 0.42 fold), PA (5.67 ± 0.56 fold), and OA (2.12 ± 0.19 fold) alone (*p* ≤ 0.0013) (Figure 1A).

As anticipated, TNF-α secretory protein levels were also higher in THP-1 cells that were co-stimulated with LPS+TG (2773.0 ± 38.92 pg/mL) and PA+TG (88.14 ± 1.63 pg/mL), compared, respectively, with LPS (922.4 ± 17.38 pg/mL) and PA (13.29 ± 2.31 pg/mL) (*p* ≤ 0.0484). Nonetheless, the OA+TG co-stimulation did not induce significantly high levels of TNF-α secretory protein, compared with OA stimulation alone (*p* = 0.0783) (Figure 1B).

### 2.2. Metabolic and/or ER Stress(es) Induce(s) the Reactive Oxygen Species (ROS)

The ER plays a significant role in oxidative stress-induced response modulation in different cells and tissues [29]. To investigate whether the ER and metabolic stresses contribute to ROS induction, ROS levels were measured in THP-1 cells that were treated with LPS, PA, and OA, alone or with ER stressor, TG. As shown via flow cytometry, higher ROS levels were co-induced via ER/metabolic stresses involving LPS (MFI_LPS+TG_ = 41,392 vs. MFI_LPS_ = 38,534 *p* = 0.0183; Figure 2A–C), PA (MFI_PA+TG_ = 19,584 vs. MFI_PA_ = 13,943 *p* ˂ 0.0001; Figure 2D–F), and OA (MFI_OA+TG_ = 17,844 vs. MFI_OA_ = 12,983 *p* = 0.0001; Figure 2G–I). Overall, LPS treatments (LPS alone and LPS+TG, Figure 2C) induced more ROS than did PA (PA alone and PA+TG, Figure 2F) and OA (OA alone and OA+TG, Figure 2I). However, comparing the normalized data, i.e., ROS ratios for treatments with/without TG, higher ROS changes were induced by PA (1.41 ± 0.01 fold increase, *p* = 0.007) and OA (1.38 ± 0.08 fold increase, *p* = 0.011), compared with LPS (1.08 ± 0.03 fold increase) (Figure 2J). Overall, these data represent the differential patterns of ROS induction in THP-1 cells in response to the ER and/or metabolic stress(es). In addition, to show that ROS expression represented the oxidation-dependent changes and not the effect of the influx/efflux of the DCFH probe by high lipid load, we treated the cells with an antioxidant curcumin before PA and PA+TG stimulations and, as expected, these data show that the pre-treatment with curcumin significantly suppressed the expression of ROS (Appendix A).

### 2.3. Lipotoxic Treatments Induce the ER Stress in THP-1 Cells

The ER plays a critical role in cellular nutrient sensing and ER stress may act as a trigger of chronic low-grade inflammation in metabolic syndromes [30]. Next, we wanted to test our hypothesis that metabolic insults induced or elevated the ER stress in THP-1 cells. To this end, we measured expression of the ER stress sensor transcripts including CCAAT-enhancer-binding protein (C/EBP) homologous protein (CHOP), activating transcription factor (ATF)-6, and inositol-requiring enzyme (IRE)-1α, also known as ER to nucleus signaling (ERN)-1, following cell stimulations with LPS, PA, and OA, alone or in the presence of TG.

The data show significant gene upregulation of *CHOP* transcripts in response to treatments with PA (2.66 ± 0.10 fold, *p* = 0.02), TG (9.76 ± 0.58 fold, *p* < 0.0001), LPS+TG (3.02 ± 0.07 fold, *p* = 0.007), PA+TG (12.25 ± 0.09 fold, *p* < 0.0001), and OA+TG (7.48 ± 0.40 fold, *p* < 0.0001), compared with control, while only the PA+TG co-treatment induced higher *CHOP* expression than that induced by TG alone (*p* = 0.002) (Figure 3A). Similarly, *ATF6* expression was induced by treatments with LPS (1.52 ± 0.06 fold, *p* = 0.02), TG (1.60 ± 0.03 fold, *p* = 0.007), LPS+TG (3.43 ± 0.19 fold, *p* < 0.0001), and PA+TG (3.03 ± 0.02 fold *p* < 0.0001), compared with control; while both LPS+TG and PA+TG co-stimulations induced higher *ATF6* expression than that induced by TG alone (*p* ˂ 0.0001) (Figure 3B). *IRE1α* expression was induced by stimulation with OA (3.44 ± 0.50 fold, *p* = 0.007) and PA+TG (2.54 ± 0.02 fold, *p* = 0.05), compared with control. However, the OA+TG co-stimulation failed to induce a higher *IRE1α* expression than that induced by TG alone (Figure 3C). Taken together, PA+TG stimulation upregulates the multiple pathways of ER stress in THP-1 cells.

### 2.4. Metabolic and ER Stresses Trigger the Cellular Antioxidant Defense Mechanisms

Induction of oxidative stress is paralleled by activation of the cellular antioxidant defense mechanisms as part of internal redox homeostasis which is critical to cellular viability, activation, proliferation, and organ function as a whole [14]. To this effect, we measured transcripts expression of superoxide dismutase (*SOD*)-*2* and nuclear factor erythroid 2-related factor (*NRF*)-*2* in THP-1 cells after metabolic stress challenge with or without TG and we found the increased *SOD2* mRNA expression following stimulations with LPS+TG (96.07 ± 3.38 fold), PA+TG (117.20 ± 4.64 fold), and OA+TG (22.47 ± 1.26 fold), compared with respective treatments without TG (*p* ˂ 0.0001) (Figure 4A). We also found the elevated *NRF2* mRNA expression in cells that were stimulated with LPS+TG (4.66 ± 0.07 fold) and PA+TG (4.00 ± 0.03 fold), compared with respective treatments without TG (*p* ˂ 0.0001) (Figure 4B). In addition, a strong correlation was found between *SOD2* and *NRF2* transcripts expression in THP-1 cells (r = 0.91 *p* ˂ 0.0001) (Figure 4C). Transcriptional data support that the cellular antioxidant defense mechanisms respond to the ER and metabolic stresses in THP-1 cells. Moreover, at the translational level, as shown by Western blots (Figure 4D), increased protein expression of SOD2 and NRF2 was induced (*p* ≤ 0.0016) by PA+TG stimulations, compared with that induced by PA stimulation alone (Figure 4E,F).

### 2.5. Metabolic/ER Stresses Induce HIF-1α Expression and MAPK/NF-κB Phosphorylation; While the Inhibitors of Inflammatory and Oxidative Pathways Suppress TNF-α Production

HIF-1 is a key transcription factor that regulates the cellular responses to oxidative and ER stresses, as part of mechanisms that drive integration of the inflammatory and metabolic responses in immune cells [31]; thus, supporting the notion that oxidative and ER stresses and inflammatory signaling are closely intertwined pathophysiological events that cross-regulate each other [32]. Therefore, we tested the stabilization of HIF-1α as well as the phosphorylation of p38, ERK1/2, and NF-κB inflammatory transcription factors in THP-1 cells that were stimulated with LPS, PA, and OA, in the presence or absence of TG as an ER stressor.

Western blot data analysis revealed that, compared with respective controls, THP-1 cell stimulation with LPS+TG and PA+TG significantly increased HIF-1α stabilization (LPS+TG: 5.31 ± 0.05 fold, PA+TG: 15.37 ± 0.10 fold) (Figure 5A,B) (*p* ˂ 0.0001) as well as induced phosphorylation of p38 (LPS+TG: 3.60 ± 0.03 fold, PA+TG: 3.61 ± 0.06 fold) (Figure 5C,D) (*p* ˂ 0.0001) and ERK1/2 (LPS+TG: 1.68 ± 0.01 fold, PA+TG: 1.94 ± 0.02 fold) (Figure 5E,F) (*p* ≤ 0.0007). More NF-κB phosphorylation was induced by the combined stimulations including LPS+TG (11.84 ± 0.04 fold), PA+TG (11.75 ± 0.05 fold), and OA+TG (5.60 ± 0.08 fold), as compared with respective solitary stimulations without TG (Figure 5G,H) (*p* ˂ 0.0001).

The immunoblot data were further validated through experiments involving MAPK and NF-κB pathway inhibitors. To this end, THP-1 cells were treated separately with U0126 and SP600125 (pharmacologic inhibitors of MAPK pathway) or with NDGA and triptolide (pharmacologic inhibitors of NF-κB pathway), before inducing metabolic and ER stresses.

As expected, inhibition of the MAPK- or NF-κB-mediated signaling induced immune reprogramming in THP-1 cells and led to a significant reduction in *TNF-α* mRNA (*p* ≤ 0.01) and protein (*p* ≤ 0.05) expression as the cells were primed with inhibitors of MAPK (Figure 6A,B) and NF-κB (Figure 6C,D) pathways prior to designated stimulations with or without TG, compared with respective control for each treatment that was likewise stimulated without involving a pathway inhibitor (instead it was primed with 0.1% BSA). However, neither of the two NF-κB pathway inhibitors induced a significant suppression in TNF-α secretory protein in response to OA+TG co-stimulation (Figure 6D).

Next, we asked how much the alleviation of oxidative stress impacted the expression of TNF-α in response to metabolic and ER stresses. To test this, THP-1 cells were first treated with antioxidants or ROS scavengers such as allopurinol, apocynin, and curcumin, and then the cells were stimulated with LPS, PA, or OA, in presence or absence of ER stressor TG. Our data show that priming with these antioxidants/ROS scavengers led to a significantly reduced TNF-α production (*p* ≤ 0.05) in response to stimulations with LPS+TG (allopurinol: 765.0 ± 35.0 pg/mL, apocynin: 446.0 ± 59.0 pg/mL, and curcumin: 62.0 ± 12.0 pg/mL), PA+TG (allopurinol: 65.0 ± 5.0 pg/mL, apocynin: 65.50 ± 7.50 pg/mL, and curcumin: 13.50 ± 1.50 pg/mL), and OA+TG (allopurinol: 11.0 ± 1.0 pg/mL, apocynin: 2.50 ± 0.50 pg/mL, and curcumin: 5.0 ± 1.0 pg/mL), compared with respective controls that were likewise stimulated following priming with 0.1% BSA as vehicle (LPS+TG: 2450.0 ± 150.0 pg/mL, PA+TG: 170.0 ± 10.0 pg/mL, and OA+TG: 52.50 ± 7.50 pg/mL). A similar trend was observed regarding individual stimulations and with TG alone (Figure 6E). However, TNF-α suppressive effects of these antioxidants were observed at the translational level while no such effect was observed at the transcriptional level (Appendix A).

Regarding whether these stresses affected the glucose uptake in THP-1 cells, we observed that both basal (non-insulin stimulated) and insulin-stimulated glucose uptake were higher in controls, compared with treated cells. Insulin-stimulated glucose uptake was significantly higher (*p* < 0.05) than non-insulin-stimulated glucose uptake in the cells that were stimulated with vehicle (0.1% BSA) or with PA+TG. However, the stimulation indices (SI) were comparable for all treatments versus control, implying that metabolic/ER stresses did not affect the glucose uptake in THP-1 cells (Appendix A).

### 2.6. Expression of TNF-α and ER Stress in Primary Monocytes from Lean and Overweight/Obese Individuals

We asked whether the metabolic (lipotoxic) and ER stresses could also similarly induce the expression of TNF-α in primary monocytes. To test this, we isolated primary monocytes from three healthy lean (BMI: 23.40 ± 0.35 kg/m^2^), one overweight (BMI: 29.20 kg/m^2^), and two obese (BMI: 31.50 ± 0.1 kg/m^2^) individuals and the cells were similarly stimulated as in experiments using THP-1 cells. The data show a similar trend of TNF-α mRNA and protein expression in primary monocytes as previously observed using THP-1 cells; thus, validating the increased TNF-α transcripts and protein expression in response to co-stimulations involving lipotoxic and ER stresses, compared to respective stimulations in the absence of ER stress, in primary monocytes isolated from the peripheral blood of healthy lean (Figure 7) and overweight/obese individuals (Figure 8).

As in THP-1 cells and as expected, co-stimulations with metabolic (lipotoxic) and ER stresses led to a higher *CHOP* mRNA expression in primary monocytes both from healthy lean and overweight/obese blood donors, compared with respective control stimulations without TG (Figure 9).

Since the ER, metabolic, and oxidative stresses orchestrate changes in adipose pathobiology during metabolic disorders, we also assessed the expression of TNF-α in adipose tissue using IHC and found an elevated TNF-α expression in obese (*p* ˂ 0001) and overweight (*p* = 0.003) individuals, compared with lean individuals. Adipose TNF-α was associated positively with BMI (r = 0.76, *p* < 0.0001). Similarly, *TNF-α* transcripts expression was also upregulated in obese (*p* < 0.0001) and overweight (*p* = 0.041), compared with lean individuals, which also correlated positively with BMI (r = 0.70, *p* < 0.0001) (Appendix A). The demographic and clinical characteristics of this population (cohort 1) are shown in Appendix A.

### 2.7. Individuals with Obesity Display Increased Expression of Systemic Inflammatory and Oxidative Stress Biomarkers

We next determined levels of systemic inflammatory and oxidative stress biomarkers in cohort 2 (Appendix A). To this end, our data show that the acute phase reactant biomarker high-sensitivity C-reactive protein (hs-CRP) levels were increased in obese individuals (27.14 ± 2.03 μg/mL, *p* < 0.0001) and overweight (7.06 ± 0.64 μg/mL; *p* = 0.03), compared with their lean (2.68 ± 0.30 μg/mL) counterparts (Figure 10A); hs-CRP levels were positively associated with BMI (r = 0.91, *p* < 0.0001) (Figure 10B). Moreover, obese individuals had increased circulatory levels of lipid peroxidation marker malondialdehyde (MDA) (9.86 ± 0.62 μg/mL) compared with both lean (4.34 ± 0.34 μg/mL; *p* < 0.0001) and overweight (6.30 ± 0.31 μg/mL; *p* < 0.0001) (Figure 10C); as well as increased levels of oxidized low-density lipoprotein (OX-LDL) (37.16 ± 1.75 U/L), compared with lean (24.98 ± 1.68 U/L; *p* < 0.0001) and overweight (31.45 ± 1.04 U/L; *p* = 0.03) individuals (Figure 10D). Notably, BMI was positively correlated with these elevations in MDA (r = 0.76, *p* < 0.0001) (Figure 10E) and OX-LDL (r = 0.77, *p* = 0.0002) (Figure 10F). Taken together, these data suggest that both systemic inflammatory and oxidative stress responses are elevated in obesity, which is in agreement with the increased systemic TNF-α levels in obese individuals (14.94 ± 2.49 pg/mL), compared with their lean (7.35 ± 1.43 pg/mL) and overweight (8.55 ± 2.53 pg/mL) counterparts (Appendix A).

## 3. Discussion

ER stress is now emerging as a critical player in inflammation and metabolic dysregulation; however, the underlying mechanisms remain elusive. We report herein that ER stress further promotes the expression of TNF-α at the transcriptional and translational levels in both THP-1 cells and primary human monocytes metabolically challenged with endotoxin (LPS), and saturated (PA) or unsaturated (OA) free fatty acids. In obesity and metabolic syndrome, a high fat diet intake leads to increased circulatory levels of LPS, contributed mainly via Gram-negative bacteria in the gut, that may cause metabolic endotoxemia and associated complications [33]. Circulatory free fatty acid levels are also high in obesity due to continued lipolysis in adipose tissue which contributes to adipose inflammation, insulin resistance, and T2D [34]. As per our findings, ER stress may act as a positive modulator of TNF-α expression in THP-1 cells and primary human monocytes, implying that the ER could also play a role as an inflammatory signaling organelle in these cells. In agreement with these observations, the ER was found to be linked with TNF-α induction via the membrane death receptor pathway and IRE1α-mediated NF-κB activation [35]. Interestingly, Nakagawa et al. showed that ER stress cooperated with hypernutrition to trigger hepatocellular carcinoma in mice via the mechanism involving TNF-α production through inflammatory macrophages in the liver [36]. Not surprisingly, the ER stress responses are now being recognized as key players in a variety of inflammatory, metabolic, and autoimmune diseases.

We next show that both metabolic and ER stresses can contribute to ROS production; however, ROS expression was significantly higher in THP-1 cells that were co-exposed to both of these cellular stresses. In our study, ER stress was induced by treating THP-1 cells or primary human monocytes with TG which is a non-competitive inhibitor of SERCA. It raises intracellular calcium levels by interfering with calcium influx into the sarcoplasmic and endoplasmic reticula and disrupts the ER calcium homeostasis, resulting in ER stress and activation of the unfolded protein response (UPR) pathways [37]. Notably, ~25% ROS are generated via disulfide bonds in the ER during the process of oxidative protein folding [38]. Accumulating evidence supports that ER intracellular calcium changes are tightly linked with ROS generation during which ER calcium outflow to mitochondria drives the ATP synthesis and mitochondrial ROS generation, in turn, leads to further calcium release [39,40].

We found that co-inducing ER and metabolic stresses in THP-1 cells and/or primary monocytes promoted the gene expression of several ER stress markers such as *CHOP* and *ATF6*. It is noteworthy that under normal conditions, *CHOP* is ubiquitously expressed at low levels; however, during ER stress, *CHOP* expression is activated in many cell types via the upstream regulatory pathway mechanism involving ATF6 translocation to the Golgi apparatus for proteolytic activation, leading to transcriptional upregulation of *CHOP* [41]. In addition to ATF6, IRE1α is another upstream stress sensor that regulates CHOP activity.

It was interesting to further note that cellular stress activated the antioxidant defense mechanisms, as was evident by increased gene and overall stable protein expression of *SOD2* and *NRF2* in THP-1 cells that were co-treated with metabolic (LPS/PA) and ER (TG) stressors. *SOD2* encodes the mitochondrial SOD2 protein, also called MnSOD, a member of the Fe^++^/Mn^++^ SOD family that transforms highly toxic oxidative phosphorylation byproduct superoxide anion (O_2_^•−^) into relatively less toxic H_2_O_2_ and molecular oxygen (O_2_), a chemical reaction that efficiently catalyzes disproportionation of O_2_^•−^ and dampens the O_2_^•−^-associated stress. As a major mitochondrial antioxidant enzyme and part of the first-line antioxidant defense, SOD2 is a key component of the complex antioxidant defense grid which alleviates oxidative stress [42]. Another major player is NRF2 which relates to the cap-n-collar (CNC) subfamily of basic region leucine zipper (bZip) transcription factors and is regarded as a master transcriptional regulator of several important genes encoding for antioxidant enzymes [43]. Our data support a strong correlation between the transcriptional upregulation of *SOD2* and *NRF2* in THP-1 cells. Consistent with our and others’ observations, the link between ER stress and the elevated expression of ROS (O_2_^•−^), SOD2, and NRF2 points to the mechanisms that maintain the ER and mitochondrial function and homeostasis [44]. Together, cumulative evidence supports that metabolic/ER stresses are linked to ROS dependent activation of the SOD2/NRF2 protective signaling in monocytic and other cells.

In addition, we found that changes in SOD2 and NRF2 mRNA and protein levels were variably expressed. Notably, the discrepancies between mRNA and protein expression may typically be attributed to varying levels of regulation between transcript and its protein product, such as differential rates of transcription, translation, turnover, and degradation; protein-per-mRNA ratios and steady-state levels; mRNA length, concentration and stability; miRNAs; translational efficiency; protein stability; the rates of lysosomal/proteasomal degradation, etc. Inter-correlations between variables and the underlying causality between measures tend to be highly complex, as supported by several studies [45,46]. In an elegant review, Maier et al. elucidated that parameters influencing mRNA–protein correlation included the mRNA abundance, RNA secondary structure, regulatory sRNAs, and regulatory proteins as translational efficiency modulators, codon bias, ribosomal density and ribosome occupancy, protein abundance and turnover, protein half-lives, experimental error and noise, and other factors such as untranslated RNA species, mRNA distribution, and nuclear sequestration [47]. Overall, SOD2/NRF2 protein expression data resonate with the sustained antioxidant defense in monocytic cells challenged with metabolic/ER stress(es).

Regarding the underlying molecular mechanisms, our data support that co-induction of metabolic/ER stresses resulted in HIF-1α stabilization and phosphorylation of the p38 MAPK- and NF-κB. Involvement of MAPK/NF-κB dependent signaling was further validated using inhibitors of MAPK (U0126 and SP600125) and NF-κB (NDGA and triptolide). TNF-α was effectively suppressed in THP-1 cells that were exposed to metabolic and ER stresses following pre-treatments with these pathway inhibitors, suggesting that p38 MAPK and p65 NF-κB are key signaling molecules involved in TNF-α expression in THP-1 cells exposed to metabolic/ER stresses. Activation of these pathways via LPS and PA or via oxidative stress is further corroborated [48,49,50,51].

Interestingly, we also noted that co-stimulation with OA and TG led to only the NF-κB phosphorylation, while stimulation with OA alone did not induce significant phosphorylation, compared with control. Consistent with this, Lamers et al., while studying the differential impact of oleate, palmitate, and adipokines on the expression of NF-κB target genes, found that stimulation with OA alone did not induce significant p65 phosphorylation in human vascular smooth muscle cells [52]. These observations are also in line with the anti-inflammatory role of oleic acid in suppressing saturated fatty acid (stearic acid)-induced proinflammatory responses in human aortic endothelial cells [53].

Of note, as observed with MAPK/NF-κB pathway inhibitors, THP-1 cells priming with antioxidants or ROS scavengers including allopurinol, apocynin, and curcumin also suppressed the production of TNF-α. This finding underscores the key role of ROS in driving TNF-α expression in THP-1 cells. Allopurinol—a xanthine oxidase inhibitor, apocynin—the NADPH oxidase inhibitor, and curcumin, also called diferuloylmethane—a principal curcuminoid from turmeric (*Curcuma longa*) and an inhibitor of cyclooxygenases; all of which act as ROS scavengers to relieve oxidative stress and alleviate inflammation via TNF-α suppression, whether used in vivo or ex vivo in various cell models [54,55,56]. It implies that the interventions aiming at ROS scavenging could have therapeutic effects in inflammatory conditions involving cellular and ER stresses driving the expression of TNF-α.

Glucose is the most important metabolic fuel to satisfy the bioenergetic needs of cells. Accordingly, we assessed the insulin-stimulated glucose uptake in THP-1 cells and found that stimulation indices for the treatments differed non-significantly from control, suggesting that TNF-α production by metabolic and ER stress stimuli was not confounded by defects in glucose uptake in these cells.

In relation to clinical aspects, we found that primary human monocytes isolated from the peripheral blood of lean and overweight/obese donors responded similarly to metabolic and ER stresses for TNF-α expression as did THP-1 cells, which supports the generalizability of our data using THP-1 cells. However, regarding the adipose compartment, we found that TNF-α protein/gene expression was higher in overweight/obese individuals compared with lean, and these changes were associated positively with BMI, as corroborated by other studies [10,57,58]. We also found that compared with lean, obese people had increased levels of systemic inflammation (CRP) and oxidative stress (MDA and OX-LDL), which had a positive association with BMI. CRP is an acute phase plasma protein synthesized in the liver in response to inflammatory cytokines and chemokines and, therefore, it is regarded as a surrogate marker of inflammatory conditions including obesity, metabolic syndrome, and atherosclerosis [59,60,61]. Chronic low-grade inflammation and glucolipotoxicity, as observed in metabolic disorders, are closely linked with loss of redox homeostasis and ROS elevation, resulting in free radical damage to the biomolecules including lipids, proteins, and nucleic acids. MDA is the most common lipid peroxidation marker used to predict the levels of oxidative stress in meta-inflammatory conditions [62], followed by others such as OX-LDL [63,64,65]. Altogether, our experimental data suggest a cooperativity mechanism between metabolic/lipotoxic and ER stresses that drives the expression of TNF-α in THP-1 cells as shown in Figure 11.

Nonetheless, this study is limited by certain caveats. First, the number of peripheral blood donors is relatively small. Second, the cells were cultured in atmospheric oxygen (in an aerobic incubator with 5% CO_2_) and humidity, which may not truly mimic the tissue microenvironment of physioxia (5–11% O_2_) or hypoxia (˂5% O_2_); however, the study objective was to induce oxidative stress in cells via the ER stress pathway using thapsigargin. Third, we could not investigate changes in the ER stress markers at the translational level which may be important, given that in addition to effects on transcriptional activation of gene expression, the integrated stress response could involve translational/post-translational modifications in the expression of ER stress markers in stressed cells. Therefore, it would be critical to address these aspects in future investigations. In any case, further studies will be required to validate these findings and to enrich clinical perspectives by including data from larger cohorts as well as through exploring more diverse cell types.

## 4. Materials and Methods

### 4.1. THP-1 Cell Cultures, Treatments, and Isolation and Stimulation of Primary Human Monocytes

THP-1 human monocytic leukemia cell line was purchased from the American Type Culture Collection (ATCC, Manassas, VA, USA). and cells were cultured in RPMI-1640 medium (Gibco, Life Technologies, Grand Island, NY, USA) containing 10% fetal bovine serum (FBS; Gibco, Life Technologies, Grand Island, NY, USA), and 2 mM glutamine (Gibco, Invitrogen, Grand Island, NY, USA), 1 mM sodium pyruvate, 10 mM HEPES, 50 U/mL penicillin and 50 μg/mL streptomycin (Gibco, Invitrogen, Grand Island, NY, USA) [66]. Cells plated (1 × 10^6^ cells/mL/well) in triplicate wells of 12-well plates were treated with LPS (10 ng/mL), PA (200 μM), OA (200 μM), TG (1 μM), LPS+TG, PA+TG, and OA+TG, or only treated with vehicle, i.e., 0.1% bovine serum albumin (BSA) (control), followed by incubation at 37 °C in a humidified incubator (5% CO_2_) for 24 h, unless otherwise stated. Cells were harvested by centrifugation (1000 rpm, 5 min) and lysed in RLT buffer (Cat. #1015762; Qiagen, GmbH, Hilden, Germany) for total RNA extraction (RNeasy kit; Qiagen, Germantown, MD, USA) and in RIPA lysis buffer (Cat. #9803; Cell Signaling Technology, Beverly, MA, USA) for total protein extraction, following the manufacturer’s instructions.

In assays involving signaling pathway inhibitors, THP-1 cells seeded in 12-well plates (1 × 10^6^ cells/mL/well) in serum-free RPMI medium were incubated for 2 h with pharmacological inhibitors of MAPK, U0126 (10 µM; Cat. #tlrl-u0126; InvivoGen, San Diego, CA, USA) and SP600125 (10 µM; Cat. #420119; Sigma-Aldrich, Merck KgaA, Darmstadt, Germany) and NF-κB inhibitors, NDGA (10 μM; Cas. #500-38-9; Sigma-Aldrich, St. Louis, MO, USA) and Triptolide (5 μM; Cas. #38748-32-2; Sigma-Aldrich, St. Louis, MO, USA) in designated triplicate wells, and then treated with metabolic and ER stressors as described above. Notably, in the stimulation control wells, cells cultured in complete RPMI medium were either left untreated (blank) or were separately pre-treated with the pathway inhibitors used, followed by stimulation of all cells with the vehicle (0.1% BSA) only. For all stimulations, the inhibitor control wells were pre-treated with vehicle (0.1% BSA) and then stimulated in the same way as other cells that were pre-treated with pathway inhibitors used. Cells were harvested by centrifugation and lyzed in RLT buffer (Cat. #1015762; Qiagen, GmbH, Hilden, Germany) for total RNA extraction using RNeasy kit (Cat. #74106; Qiagen, Germantown, MD, USA), following the manufacturer’s instructions, while cell supernatants were aliquoted and stored at −80 °C for measuring TNF-α concentrations via ELISA.

In assays involving ROS scavengers or antioxidants, THP-1 cells dispensed in triplicate wells of 12-well plates (1 × 10^6^ cells/mL/well) were pre-incubated for 1 h with allopurinol (100 μM; Cas. #315-30-0; Sigma-Aldrich, St. Louis, MO, USA), apocynin (100 μM; Cas. #498-02-2; Sigma-Aldrich, St. Louis, MO, USA), and curcumin (10 μM; Cas. #458-37-7; Sigma-Aldrich, St. Louis, MO, USA) in designated wells, followed by stimulations with metabolic and/or ER stressors as described before. Notably, antioxidant control wells were pre-treated with vehicle (0.1% BSA) and then stimulated in the same way as other cells that were pre-treated with antioxidants. After 24 h incubation, cell supernatants were collected, aliquoted, and stored at −80 °C for measuring TNF-α concentrations via ELISA.

The peripheral blood from six healthy adult male donors including three lean (BMI: 23.40 ± 0.35 kg/m^2^), one overweight (BMI: 29.20 kg/m^2^), and two individuals with obesity (BMI: 31.50 ± 0.1 kg/m^2^) was collected in EDTA vacutainer tubes by the phlebotomy staff of the Dasman Diabetes Institute, Kuwait, per the informed consent of donors before sample collection. Blood collection and processing were in agreement with the principles and ethics of the Declaration of Helsinki 1975 (most recent iteration in 2013) and protocol approval by the institutional ethics committee (Ref. #RA-HM-2019-030 KADEM). Peripheral blood mononuclear cells (PBMCs) were isolated using the Ficoll–Hypaque density gradient method described elsewhere [67]. Primary monocytes were isolated from PBMCs as previously described [68]. Briefly, PBMCs were dispensed in 6-well plates (Costar; Corning Optical Communications GmbH & Co. KG, UAE) at a cell density of 2 × 10^6^ cells in 2 mL per well and cultured in serum-free medium at 37 °C for 6 h. Non-adherent cells were flushed off using fresh serum-free medium and the adherent cells were further incubated at 37 °C for 24 h in RPMI complete medium containing 10% FBS. Cells were stimulated with LPS (10 ng/mL), PA (200 μM), OA (200 μM), TG (1 μM), LPS+TG, PA+TG, and OA+TG, or only treated with 0.1% BSA as vehicle control, followed by incubation at 37 °C for 24 h. At the end of incubation, cells were collected for total RNA extraction and supernatants were collected for measuring TNF-α secreted protein levels.

### 4.2. Real-Time Quantitative Reverse Transcription (qRT)-PCR

Total RNA was extracted using an RNeasy kit (Qiagen, Germantown, MD, USA) and following the manufacturer’s instructions. RNA was quantified (Epoch™ Spectrophotometer System; BioTek, Santa Clara, CA, USA) and 1 μg RNA sample was used to prepare cDNA using TaqMan reagents (high capacity cDNA reverse transcription kit; Applied Biosystems, Foster City, CA, USA) [69]. For real-time RT-PCR, a 50 ng cDNA sample was amplified using TaqMan^®^ Gene Expression Master Mix (Cat. #4369016; Applied Biosystems, Foster City, CA, USA) and TaqMan Gene Expression Assay (Cat. #4331182; Applied Biosystems, Foster City, CA, USA), using target gene-specific products (Cat. #4331182; ThermoFisher Sci.) including TNF-α (Hs00174128_m1), CHOP (Hs00358796_g1), ATF6 (Hs00232586_m1), IRE1α (Hs00980095_m1), SOD2 (Hs00167309_m1), NRF2 (Hs00202227_m1), and GAPDH (Hs02786624_g1) containing forward and reverse gene specific primers and a target specific TaqMan^®^ 5′-FAM-labeled and 3′-NFQ-labeled MGB probe, using 40 PCR amplification cycles in 7500 Fast Real-Time PCR System (Applied Biosystems, CA, USA). Each cycle was as follows: denaturation (95 °C, 15 s) and annealing/extension (60 °C, 1 min), following activation of the uracil DNA glycosylase (UDG) (50 °C, 2 min) and AmpliTaq Gold enzyme (95 °C, 10 min). Target gene expression relative to that in the control sample was calculated by comparative 2^−ΔΔCT^ method; data were normalized to GAPDH expression and expressed as fold change over average gene expression in the control sample taken as 1.

### 4.3. Enzyme-Linked Immunosorbent Assays (ELISAs)

TNF-α ELISA was performed following the manufacturer’s instructions (Human DuoSet TNF-α ELISA kit, Cat. #DY210-5; R&D Systems Inc., Minneapolis, MN, USA). Briefly, cell supernatants and standards were added (100 μL/well) to designated triplicate wells and incubated at room temperature (RT) for 2 h. After aspiration/washing 3 times, the detection antibody was added (100 μL/well), and incubated at RT for 2 h. After 3 washes as before, a working dilution of streptavidin-HRP was added (100 μL/well) and incubated at RT in the dark for 20 min. After 3 washes, substrate was added (100 μL/well) and incubated at RT in the dark for 20 min. Lastly, a stop solution was added (50 μL/well), optical density (O.D.) was read at 450 nm wavelength (corrections at 540 nm or 570 nm), and TNF-α concentrations in cell supernatants were calculated from the standard curve.

To perform high sensitivity C-reactive protein (hs-CRP) ELISA (Quantikine C-Reactive Protein/CRP ELISA kit, Cat. #DCRP00; R&D Systems Inc. Minneapolis, MN, USA), standards, controls, and diluted plasma samples were added (50 μL/well) to designated triplicate wells containing assay diluent (100 μL/well) and incubated (RT, 2 h). After 4 washes, human CRP conjugate was added (200 μL/well) and incubated as before. After 4 washes, substrate solution was added (200 μL/well) and incubated in the dark (RT, 30 min). Finally, a stop solution was added (50 μL/well) and O.D. was read at 450 nm within 30 min (corrections at 540 nm or 570 nm). hs-CRP concentrations were calculated from the standard curve and adjusted for the dilution factor.

For malondialdehyde (MDA) ELISA (Human Malondialdehyde/MDA ELISA kit, Cat #LS-F4236; LS Bio, Shirley, MA, USA), standards, blanks, and plasma samples were added (50 μL/well) to designated triplicate wells, detection reagent A was added immediately to each well (50 μL/well), incubated (37 °C, 1 h), washed 3 times, then detection reagent B was added to each well (100 μL/well) and again incubated (37 °C, 30 min). After 5 washes, TMB substrate solution was added to each well (90 μL/well) and incubated in the dark (37 °C) for 15 min. Lastly, a stop solution was added to each well (50 μL/well), O.D. was read immediately at 450 nm, and MDA concentrations were calculated from standard curve.

To carry out oxidized low-density lipoprotein (OX-LDL) ELISA (Human Oxidized LDL ELISA kit, Cat #10-114301; Mercodia, Uppsala, Sweden), calibrators, controls, and diluted plasma samples were added (25 μL/well) to designated triplicate wells containing assay diluent (200 μL/well) and incubated on a plate shaker (700–900 rpm, RT) for 2 h. After 6 washes, enzyme conjugate was added (200 μL/well) and incubated under shaking for 1 h as before. After 6 washes, TMB substrate was added (200 μL/well) and incubated without shaking at RT for 15 min. Finally, stop solution was added (50 μL/well), O.D. was read at 450 nm within 30 min, and OX-LDL concentration were calculated from standard curve and adjusted for the sample dilution factor.

### 4.4. ROS Detection Assay

To measure intracellular ROS, THP-1 cells were stimulated with LPS (10 ng/mL), PA (200 μM), and OA (200 μM), alone or with TG (1 μM), and control was treated with vehicle (0.1% BSA) only. After incubation at 37 °C in a humidified incubator (5% CO_2_) for 24 h, ROS induction was measured using ROS assay kit (Cat. #KP-06-003 BQC Kit; BioQueChem Inc., Llanera-Asturias, Spain), based on uptake of cell-permeant fluorogenic probe 2′-7′dichlorofluorescein diacetate (DCFH-DA). Following cell incubation with the labeled probe for 15 min, DCFH-DA is hydrolyzed by cellular esterases into DCFH carboxylate which is oxidized by intracellular ROS into the fluorescent 2′-7′dichlorofluorescein (DCF) product, which is measured by flow cytometry. After the end of incubation, cells in culture media were loaded with DCFH-DA probe (15 μM), incubated at 37 °C for 30 min, and analyzed (without washing) by flow cytometry. The final DCF product was excited using a 488 nm laser and detected at 535 nm wavelength, and intracellular ROS induction was expressed as mean fluorescence intensity (MFI) values.

### 4.5. Western Blotting

Regarding immunoblotting, THP-1 cells were plated at a cell density of 1 × 10^6^ cells/mL in triplicate wells of 12-well plates and stimulated with LPS (10 ng/mL for 30 min), PA (200 μM for 2 h), and OA (200 μM for 2 h), alone or with TG (1 μM for 24 h), and control was treated with vehicle (0.1% BSA) only. Cells were lysed by incubating for 30 min with lysis buffer containing Tris 62.5 mM (pH 7.5), 1% Triton X-100, and 10% glycerol, lysates were clarified by centrifugation (14,000 rpm, 4 °C, 10 min), and supernatants were collected. Protein concentrations were measured using QuickStart Bradford Dye (Cat. #5000205; Bio-Rad Laboratories, Hercules, CA, USA) and samples were resolved by 12% sodium dodecyl–sulfate polyacrylamide gel electrophoresis (SDS-PAGE) [70]. Blots were probed overnight at 4 °C with rabbit anti-human monoclonal antibodies (Cell Signaling Technology, MA, USA; 1:1000 dilution) against the following: IRE1α (14C10) (Cat. #3294), SOD2 (D3X8F) (Cat. #13141), NRF2 (D1Z9C) (Cat. #12721), HIF-1α (Cat. #36169S), phospho-p38 MAPK (Cat. #9216), total-p38 MAPK (Cat. #9212), phospho-ERK1/2 (Cat. #9101), total-ERK1/2 (Cat. #9102), phospho-NF-κB (Cat. #3031), and total- NF-κB (Cat. #3034s). Phospho (S724)-IRE1α (Cat. #ab48187) was purchased from Abcam (Waltham, MA, USA). Blots were washed 3 times with TBS wash buffer and incubated for 2 h with HRP-conjugated secondary antibody (Cat. # W4011, anti-rabbit IgG (H+L), HRP conjugate, 1:2500 dilution; Promega, WI, USA). Bands were developed (Amersham ECL Plus Western Blot Detection System; GE HealthCare, Buckinghamshire, UK) and visualized (ImageDoc™; MP Imaging Systems, BioRad Laboratories, Hercules, CA, USA). Band densities expressed as arbitrary units (AU) were determined and data (mean ± SEM) were compared against control using statistical analysis.

### 4.6. Insulin-Stimulated Glucose Uptake Assay

To perform the insulin-stimulated glucose uptake assay, cultured THP-1 cells were plated at a cell density of 1 × 10^6^ cells/mL/well in triplicate wells of 12-well plates and cells were stimulated with LPS (10 ng/mL), PA (200 μM), and OA (200 μM), in the presence of TG (1 μM), while control was treated with vehicle (0.1% BSA) only. After incubation at 37 °C (5% CO_2_) for 24 h, cells were PBS washed 3 times, resuspended in serum-free RPMI-1640 medium, plated (1 × 10^6^ cells/mL/well), and again incubated at 37 °C (5% CO_2_) for 18 h. Glucose uptake assay was performed, following the manufacturer’s protocol (Cat. #ab136955, Colorimetric glucose uptake assay kit; Abcam, MA, USA). Briefly, cells were washed 3 times and split in three aliquots (5 × 10^4^ cells/100 μL/aliquot) for each treatment in duplicate as follows: (1) sample background control; (2) insulin stimulated cells; and (3) non-insulin stimulated control. After washing, cells were glucose-starved by pre-incubation with 100 μL KRPH buffer (2% BSA) for 40 min. Sample background control was washed 3× but 2-deoxyglucose (2-DG, a glucose analog) was not added. Insulin stimulated cells were incubated with 2 μM insulin in KRPH buffer (2% BSA) for 20 min to activate glucose transporters and then 10 mM 2-DG (10 μL) was added to both insulin stimulated cells and non-insulin stimulated control, and incubated at 37 °C for 20 min. Later, all cells were washed 3× with PBS, lysed by one freeze–thaw in extraction buffer (80 μL/sample), heated at 85 °C for 40 min, cooled on wet ice for 5 min, and 10 μL neutralizing buffer was added to each sample. Supernatants (5 μL each) were transferred to new tubes, diluted 1:10 using assay buffer, 10 μL Reaction Mix A (NADPH generation) was added to each of standard, controls, and sample tubes, and incubated at 37 °C for 1 h. Then, 90 μL extraction buffer was added to each tube, heated at 90 °C for 40 min, cooled on wet ice for 5 min, and neutralized by adding neutralizing buffer (12 μL/tube). Reaction Mix B was added (38 μL) to each tube, vortexed briefly, and samples (100 μL each) were transferred to 96-well black microplate (Cat. #CLS3603; Sigma-Aldrich Inc., St. Louis, MO, USA) and O.D. was read at 412 nm in a kinetic mode (Synergy H4 Hybrid microplate reader; BioTec, Winooski, VT, USA), every 2–3 min at 37 °C (protected from light) until the standard (#6) with the highest 2-DG6P concentration (100 pmol/well) reached O.D. 1.5–2.0. Levels of glucose uptake were calculated as follows: 2-DG uptake = (Ts/Sv) × D = pmol/μL = nmol/mL = μM; where Ts = amount of 2-DG6P in sample tube calculated from standard curve (pmol), Sv = sample volume (μL), D = sample dilution factor (if diluted for optimization).

### 4.7. Study Participants, Anthropometry, Adipose Tissue Biopsies, and Plasma Lipid Profiles

Regarding clinical investigations, cohort 1 comprising 29 and cohort 2 comprising 36 individuals were recruited in the study. Based on body mass index (BMI), participants were classified as lean (BMI 18.5 to 24.9 kg/m^2^), overweight (BMI 25 to 29.9 kg/m^2^), and obese (BMI ≥ 30 kg/m^2^). Written informed consent was obtained from each individual, following ethical guidelines of the Declaration of Helsinki and study approval by the ethics committee of the Dasman Diabetes Institute, Kuwait (Protocols #: RA 2015-027; RA 2010-003).

For anthropometric measurements, waist circumference was measured by constant tension tape, height by inflexible height measuring bars, and weight by electronic weighing scales.

Subcutaneous adipose biopsies (~500 mg) were collected from cohort 1 (characteristics are shown in Appendix A), from the periumbilical area using a standard sterile procedure [71,72]. Briefly, after skin disinfection by ethanol gauze and local anesthesia by 2% lidocaine (Fresenius Kabi, LLC, Lake Zurich, IL, USA), fat tissue was collected via a small (~0.5 cm) skin incision, divided into small fractions (~50 mg), transferred to RNAlater for preservation at −80 °C (Sigma-Aldrich Chemie GmbH, Taufkirchen, Germany) or immediately put into 10% neutral-buffered formalin for paraffin embedding.

Plasma samples from cohort 2 (characteristics are shown in Appendix A) were used to study the systemic inflammatory (hs-CRP levels) and oxidative stress (MDA and OX-LDL levels) profiles by using commercial kits as described before.

Blood glucose was detected using blood glucose and ketone monitoring system (Cat. #71373-80; Freestyle Optium Neo, Abbot, Maidenhead, Berkshire, UK) and lipid profiles were measured including total cholesterol (Cat. #ab287836, human total cholesterol ELISA kit; Abcam, Waltham, MA, USA), high-density/low-density lipoproteins (HDL/LDL; Cat. #ab65390; Cholesterol assay kit, Abcam, Waltham, MA, USA), and triglyceride (Cat. #ab65336; Triglyceride assay kit, Abcam, Waltham, MA, USA), following the recommended protocols.

Peripheral blood samples for PBMC isolation were obtained from three healthy lean individuals (23.40 ± 0.35 kg/m^2^), one overweight (BMI: 29.20 kg/m^2^), and two individuals with obesity (BMI: 31.50 ± 0.1 kg/m^2^), following written informed consent per the ethical principles of the Declaration of Helsinki (updated 2013) and study approval by the ethics committee of the Dasman Diabetes Institute, Kuwait (Protocol #: RA-HM-2019-030 KADEM). Adherent cells isolated from cultured PBMCs were further used in experiments involving primary monocyte stimulations, as described earlier for THP-1 cells.

### 4.8. Immunohistochemistry (IHC)

IHC was performed as described before [73]. Briefly, adipose tissue sections (4 μm) from paraffin-embedded samples on slides were deparaffinized by xylene and rehydrated by passing through descending ethanol grades in water (100%, 95%, and 75%). Antigen was retrieved (Target retrieval solution, pH 6.0; Dako, Glostrup, Denmark) by pressure cooker boiling (8 min) and cooling (15 min). After PBS wash (RT, 30 min), tissue slides were treated (30 min) with 3% hydrogen peroxide (H_2_O_2_) to block endogenous peroxidases and non-specific antibody staining (background) was blocked first by 5% non-fat milk and later by 1% BSA, 1h incubation each. Samples were treated (RT) overnight with primary antibody (1:800 dilution, Cat. #ab9635, rabbit anti-human TNF-α polyclonal antibody; Abcam, MA, USA). After 3 washes (PBS-0.5% Tween), slides were incubated for 1 h with secondary antibody (goat anti-rabbit, horseradish peroxidase-conjugated polymer chain, EnVision kit; Dako, Glostrup, Denmark) and color was developed using chromogenic substrate 3,3′-diaminobenzidine (DAB). Samples were thoroughly washed under running water, counterstained with Harris hematoxylin, dehydrated by passing through ascending ethanol grades in water (75%, 95%, and 100%), and mounted in dibutylphthalate xylene (DPX). For analysis, digital photomicrographs (20× magnification) of at least 4 different regions were taken to assess the regional heterogeneity in tissue sections (Panoramic Scan 3DHistech, Budapest, Hungary). Samples were analyzed and compared using ImageJ (1.5.3 version) (NIH, Bethesda, MD, USA), as per the provided instructions.

### 4.9. Statistical Analysis

The data were expressed as mean ± SEM values and group means were compared using one-way or two-way ANOVA, Dunnett’s/Tukey’s, or Sidak’s multiple comparisons tests as appropriate. Spearman correlation analysis was performed to determine associations between different variables. GraphPad Prism 9.4.1.681 (GraphPad Software, San Diego, CA, USA) was used for statistical analysis of the data and to prepare graphs. All *p*-values ≤ 0.05 were considered significant, and the statistical significance was expressed as * *p* ˂ 0.05, ** *p* ˂ 0.01, *** *p* ˂ 0.001, and **** *p* ˂ 0.0001.

## 5. Conclusions

In conclusion, we show that a cooperative interaction between cellular stresses, including a metabolic stress challenge and ER stress, could induce intracellular ROS and promote the expression of TNF-α in THP-1 cells via the ROS/CHOP/HIF-1α and MAPK/NF-κB dependent mechanisms. A similar trend in TNF-α expression was observed in primary human monocytes derived from healthy lean persons as well as those with overweight or obesity. Individuals with obesity were found to have upregulated adipose and systemic TNF-α expression, together with increased plasma levels of hs-CRP, MDA, and OX-LDL, all of which were positively associated with BMI. Taken together, these data imply that the intertwined metabolic and ER stresses may act as potential drivers of TNF-α and as co-player in metabolic inflammation. Importantly, our study also shows that priming with curcumin, allopurinol, or apocynin suppress the TNF-α production by THP-1 cells which points to the therapeutic potential of ROS scavengers and antioxidants in inflammatory conditions involving cellular and metabolic stresses.

## Figures and Tables

**Figure 1 ijms-24-15186-f001:**
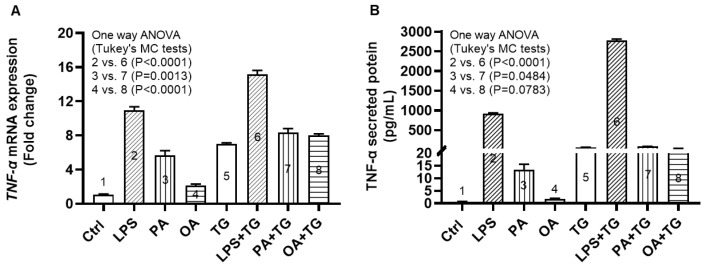
ER stress promotes the metabolic stress-induced TNF-α mRNA/protein expression in monocytic cells. THP-1 cells were seeded (1 × 10^6^ cells/mL/well) in triplicate wells of 12-well plates and treated with different metabolic stress inducers including LPS (10 ng/mL), PA (200 μM), and OA (200 μM), in presence or absence of the ER stressor thapsigargin (TG, 1 μM), while control was treated with the vehicle (0.1% BSA) only, and the cells were incubated for 24 h. Total RNA was extracted from cells for measuring *TNF-α* gene expression using qRT-PCR and cell supernatants were used to detect levels of TNF-α secreted protein via ELISA as described in Section 4. Similar results were obtained from three independent experiments. Data (expressed as mean ± SEM) were analyzed using one-way ANOVA, Tukey’s multiple comparisons test, and *p*-values ≤ 0.05 were considered significant. The representative data show that the ER stress (TG treatment) increases: (**A**) TNF-α mRNA expression in monocytic cells that were treated with LPS (bars 2 vs. 6), PA (bars 3 vs. 7), and OA (bars 4 vs. 8); and (**B**) TNF-α secreted protein levels in response to treatments with LPS (bars 2 vs. 6), PA (bars 3 vs. 7), and OA (bars 4 vs. 8). *p*-values ≤ 0.0484. However, the difference between TNF-α protein induction by OA+TG co-stimulation and OA stimulation alone did not reach statistical significance (*p* = 0.0783).

**Figure 2 ijms-24-15186-f002:**
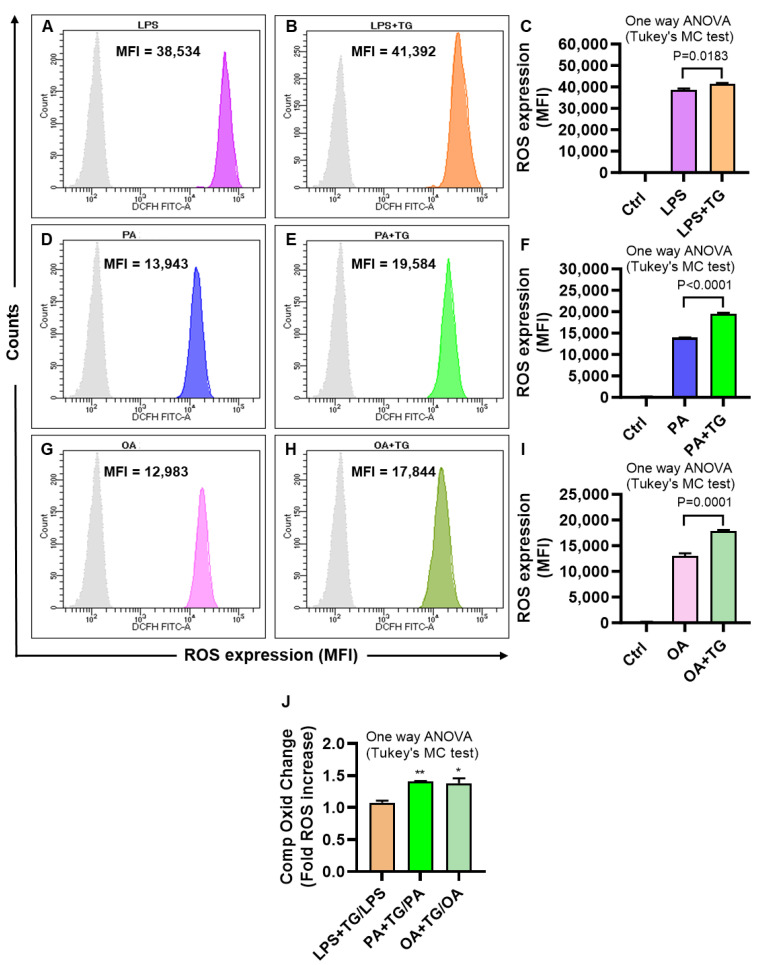
Metabolic and ER stresses induce the intracellular reactive oxygen species (ROS). THP-1 cells were plated (1 × 10^6^ cells/mL/well) in triplicate wells of 12-well plates and treated with LPS (10 ng/mL), PA (200 μM), and OA (200 μM), in presence or absence of the ER stressor thapsigargin (TG, 1 μM) while control was treated with vehicle (0.1% BSA) only, and the cells were incubated for 24 h. Intracellular ROS was measured using DCFH-DA assay and flow cytometry as described in Section 4. Similar results were obtained from three independent experiments. Data (expressed as mean ± SEM) were analyzed using one-way ANOVA, Tukey’s multiple comparisons test, and *p*-values ≤ 0.05 were considered significant. The representative data show that the ER stress (TG treatment) promotes the ROS in cells that are metabolically stressed from treatments involving: (**A**–**C**) LPS, (**D**–**F**) PA, and (**G**–**I**) OA (*p* ≤ 0.0183). The maximum ROS induction was noted for (**C**) LPS+TG treatment (MFI: 41,392 ± 527.10), followed by (**F**) PA+TG treatment (MFI: 19,584 ± 200.90) and (**I**) OA+TG treatment (MFI: 17,844 ± 243.10). (**J**) The relative ROS induction by TG, comparing the normalized ratios, was the highest for PA (1.41 ± 0.01 fold increase), followed in order by OA (1.38 ± 0.08 fold increase), and LPS (1.08 ± 0.03 fold increase). Significant inductions by PA (** *p* = 0.007) and OA (* *p* = 0.011) were observed, compared with LPS.

**Figure 3 ijms-24-15186-f003:**
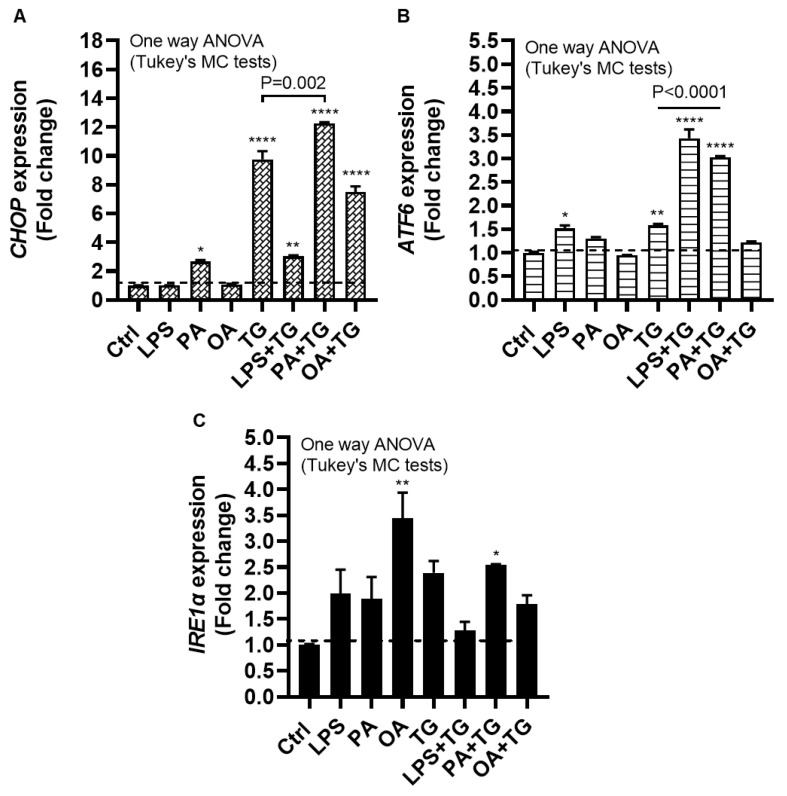
Metabolic stress induces or promotes ER stress. THP-1 cells were dispensed (1 × 10^6^ cells/mL/well) in triplicate wells of 12-well plates and treated with LPS (10 ng/mL), PA (200 μM), and OA (200 μM), in presence or absence of ER stress inducer thapsigargin (TG, 1 μM) while control was treated with the vehicle (0.1% BSA) only, and the cells were incubated for 24 h. Total RNA was extracted and the gene expression of ER stress markers including *CHOP*, *ATF6*, and *IRE1α* was determined using qRT-PCR as described in Section 4. Similar results were obtained from three independent experiments. Data (expressed as mean ± SEM) were analyzed using one-way ANOVA, Tukey’s/Dunnett’s multiple comparisons test, and *p*-values ≤ 0.05 were considered significant. The representative data show, compared with control, the increased: (**A**) *CHOP* mRNA levels in cells treated with PA, TG, LPS+TG, PA+TG, and OA+TG; (**B**) *ATF6* mRNA levels in cells treated with LPS, TG, LPS+TG, and PA+TG; and (**C**) *IRE1α* mRNA levels in cells treated with OA and PA+TG. Statistical significance is shown as * *p* < 0.05, ** *p* < 0.01, and **** *p* < 0.0001, compared with respective control (vehicle treatment).

**Figure 4 ijms-24-15186-f004:**
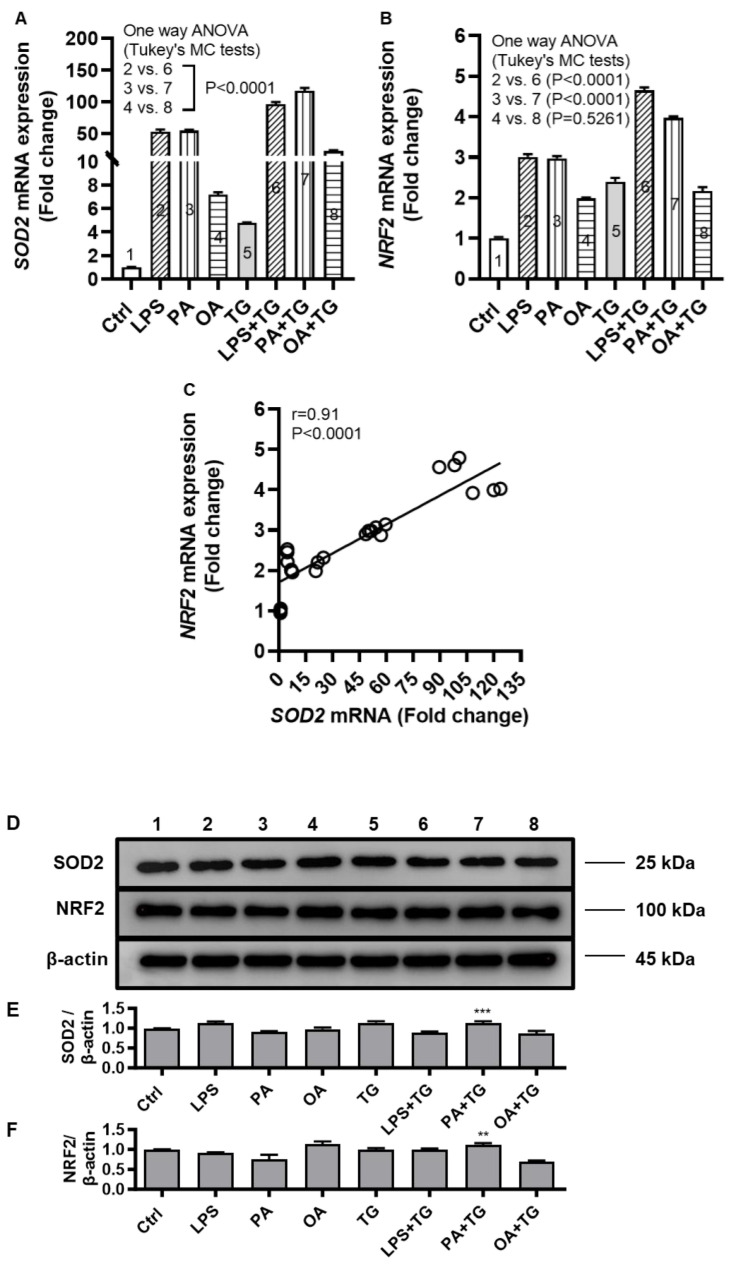
Metabolic and/or ER stress(es) activate(s) the antioxidant defense mechanisms. THP-1 cells were seeded (1 × 10^6^ cells/mL/well) in triplicate wells of 12-well plates and treated with LPS (10 ng/mL), PA (200 μM), and OA (200 μM), in presence or absence of the ER stress inducer thapsigargin (TG) (1 μM) while control (Ctrl) was treated with the vehicle (0.1% BSA) only, and the cells were incubated for 24 h. Total RNA was extracted and the gene expression of *SOD2* and *NRF2* was determined using qRT-PCR while SOD2 and NRF2 protein expression in cell lysates was assessed using Western blotting as described in Section 4. Similar results were obtained from two independent experiments. Data (expressed as mean ± SEM) were analyzed using one-way ANOVA, Tukey’s or Dunnett’s multiple comparisons test, as appropriate. All *p*-values ≤ 0.05 were considered significant. The representative data show that metabolic and/or ER stress(es) upregulate(s) the mRNA expression of (**A**) *SOD2* and (**B**) *NRF2* in monocytic cells (*p* < 0.0001); except *NRF2* expression in response to OA+TG treatment (*p* = 0.5261). (**C**) Based on gene expression data, a strong agreement was found between *SOD2* and *NRF2* (r = 0.91, *p* ˂ 0.0001). (**D**) Immunoblots show SOD2 and NRF2 expression in response to metabolic and/or ER stress treatments in THP-1 cells. (**E**,**F**) Increased expression of SOD2 (*p* = 0.0006) and NRF2 (*p* = 0.0016) was induced by THP-1 cell treatment with PA+TG, as compared to treatment with PA alone. ** *p* < 0.01 and *** *p* < 0.001.

**Figure 5 ijms-24-15186-f005:**
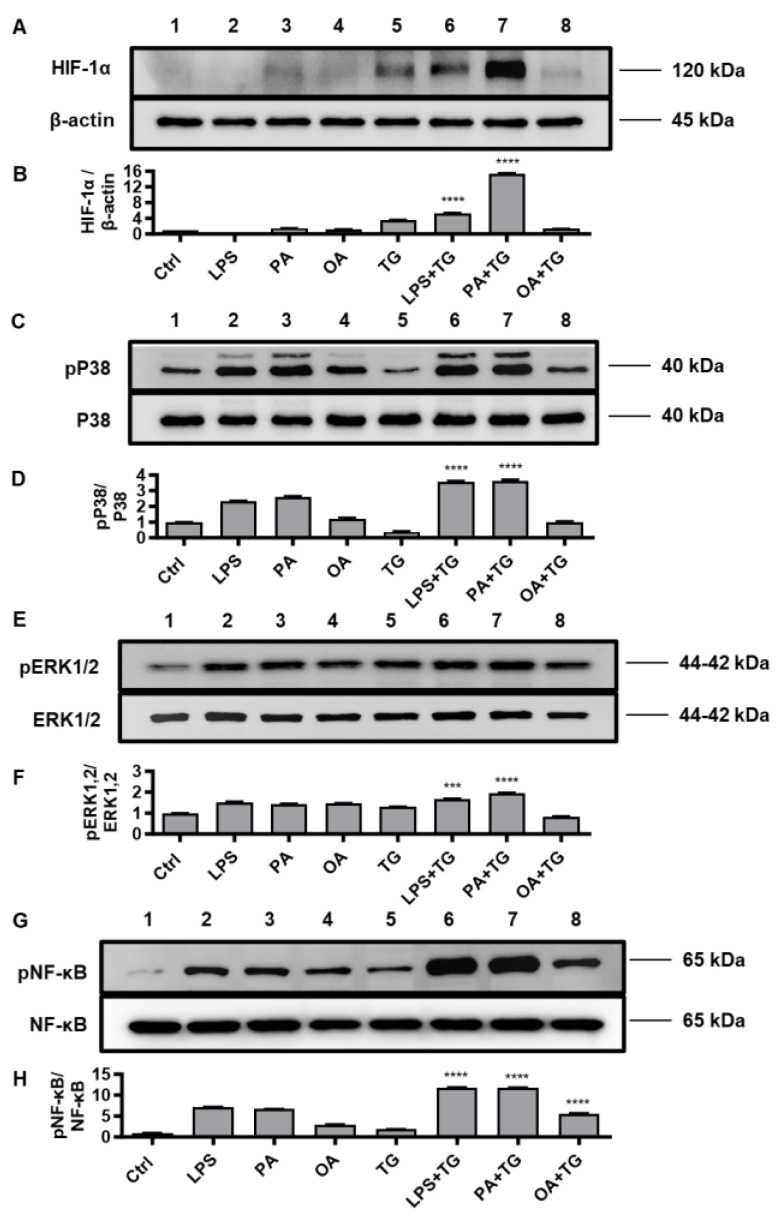
Metabolic and ER stresses co-induce stabilization of HIF-1α and phosphorylation of p38 MAPK/NFκB signaling proteins. THP-1 cells seeded at a cell density of 1 × 10^6^ cells/mL in triplicate wells of 12-well plates were treated with LPS (10 ng/mL for 30 min), PA (200 μM for 2 h), and OA (200 μM for 2 h), in presence and absence of ER stressor TG (1 μM for 24 h), while control was treated with vehicle (0.1% BSA) only. Cells were lysed in RIPA buffer for total protein extraction, resolved by 12% SDS-PAGE, and immunoblots were analyzed for the expression of HIF-1α, β-actin, phospho/total p38, phospho/total ERK1/2, and phospho/total NF-κB, as described in Section 4. Similar results were obtained from three independent experiments. Data (expressed as mean ± SEM) were analyzed using one-way ANOVA, Tukey’s multiple comparisons test, and *p*-values ≤ 0.05 were considered significant. The representative data show, compared with respective controls, increased levels of: (**A**,**B**) HIF-1α expression in cells treated with LPS+TG and PA+TG; (**C**,**D**) p38 phosphorylation in cells treated with LPS+TG and PA+TG; (**E**,**F**) ERK1/2 phosphorylation in cells treated with PA+TG; and (**G**,**H**) NF-κB phosphorylation in cells treated with LPS+TG, PA+TG, and OA+TG. Statistical significance is shown for differences, compared with respective treatment without TG. *** *p* < 0.001 and **** *p* < 0.0001.

**Figure 6 ijms-24-15186-f006:**
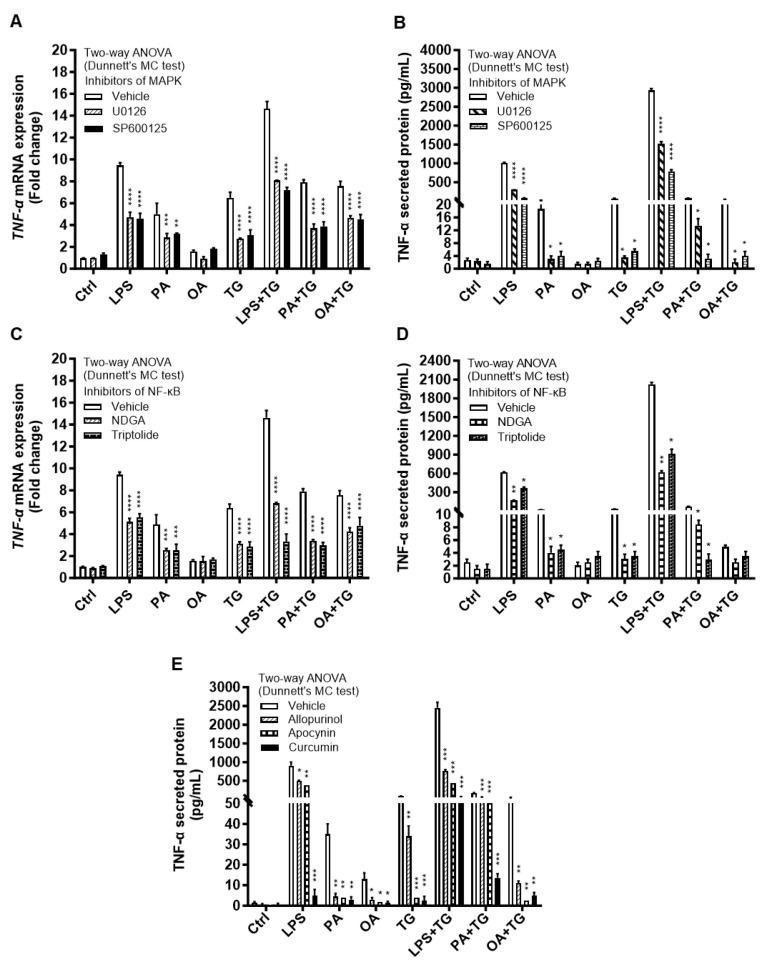
Inhibition of MAPK/NFκB-mediated signaling or ROS scavenging suppresses the TNF-α expression in THP-1 cells. The cells were plated at a cell density of 1 × 10^6^ cells/mL/well in triplicate wells of 12-well plates. Cells were pre-treated (for 2 h) with pharmacological inhibitors of MAPK (U0126 and SP600125) and NF-κB (NDGA and Triptolide) pathways, or incubated for 1 h in designated wells with allopurinol (100 μM), apocynin (100 μM), and curcumin (10 μM), followed by stimulation with LPS (10 ng/mL), PA (200 μM), and OA (200 μM), with or without ER stressor thapsigargin (TG, 1 μM). In stimulation control wells, cells were either left untreated (blank) or pre-treated (separately) with inhibitors/antioxidants, followed by stimulation of all cells using vehicle (0.1% BSA) only. In inhibitor/antioxidant control wells for each stimulation, cells were pre-treated with vehicle (0.1% BSA) and later stimulated likewise other cells that were pre-treated with pathway inhibitors or antioxidants. After 24 h incubation, total RNA was extracted for determining *TNF-α* mRNA expression via qRT-PCR and cell supernatants were analyzed for TNF-α secreted protein levels using ELISA as described in Section 4. Similar results were obtained from three independent experiments. Data (expressed as mean ± SEM) were analyzed using two-way ANOVA (Dunnett’s multiple comparisons test) and *p*-values ≤ 0.05 were considered significant. The representative data show that, compared with respective controls, inhibition of the MAPK and NF-κB pathways led to a significant suppression of TNF-α at the (**A**,**B**) transcriptional (mRNA) and (**C**,**D**) translational (protein) levels. However, inhibition of the NF-κB pathway (using NDGA and Triptolide) did not cause a significant reduction in TNF-α secretion in response to cell co-stimulation with OA+TG. (**E**) Similarly, THP-1 cell priming with antioxidants or ROS scavengers led to a significant suppression of TNF-α production by THP-1 cells. Statistical significance is shown as * *p* < 0.05, ** *p* < 0.01, *** *p* < 0.001, and **** *p* < 0.0001, compared with respective vehicle control.

**Figure 7 ijms-24-15186-f007:**
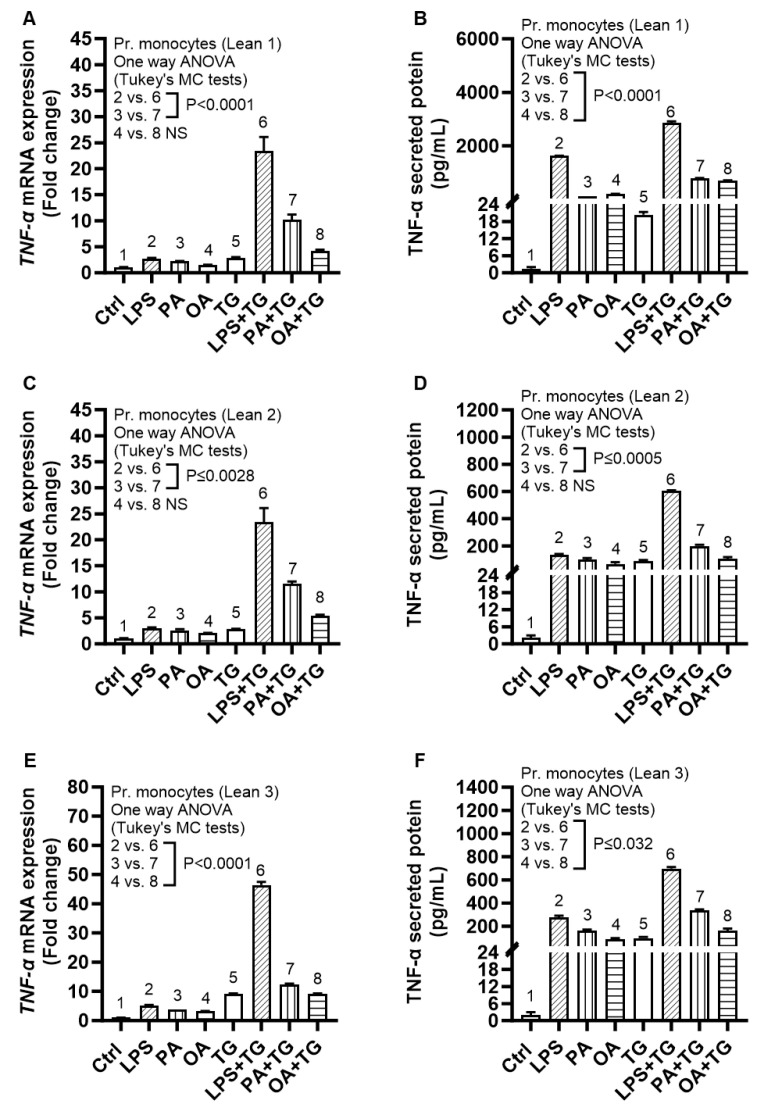
Increased TNF-α expression in primary monocytes isolated from lean individuals, following co-stimulation with metabolic (lipotoxic) and ER stresses. Primary monocytes were isolated from the peripheral blood mononuclear cells (PBMCs) of three healthy lean individuals (BMI: 23.40 ± 0.35 kg/m^2^) and the cells were stimulated with LPS (10 ng/mL), PA (200 μM), and OA (200 μM), with or without ER stressor thapsigargin (TG) (1 μM), while cells treated with vehicle only represent control (Ctrl). TNF-α gene and secreted protein expression was determined using qRT-PCR and ELISA, respectively, as described in Section 4. Similar results were obtained from three independent experiments. Data (mean ± SEM) were analyzed using one-way ANOVA and group means were compared using Tukey’s multiple comparisons test. All *p*-values ≤ 0.05 were considered significant. The representative data show increased TNF-α mRNA and secreted protein expression in samples from: (**A**,**B**) lean 1; (**C**,**D**) lean 2; and (**E**,**F**) lean 3 blood donors.

**Figure 8 ijms-24-15186-f008:**
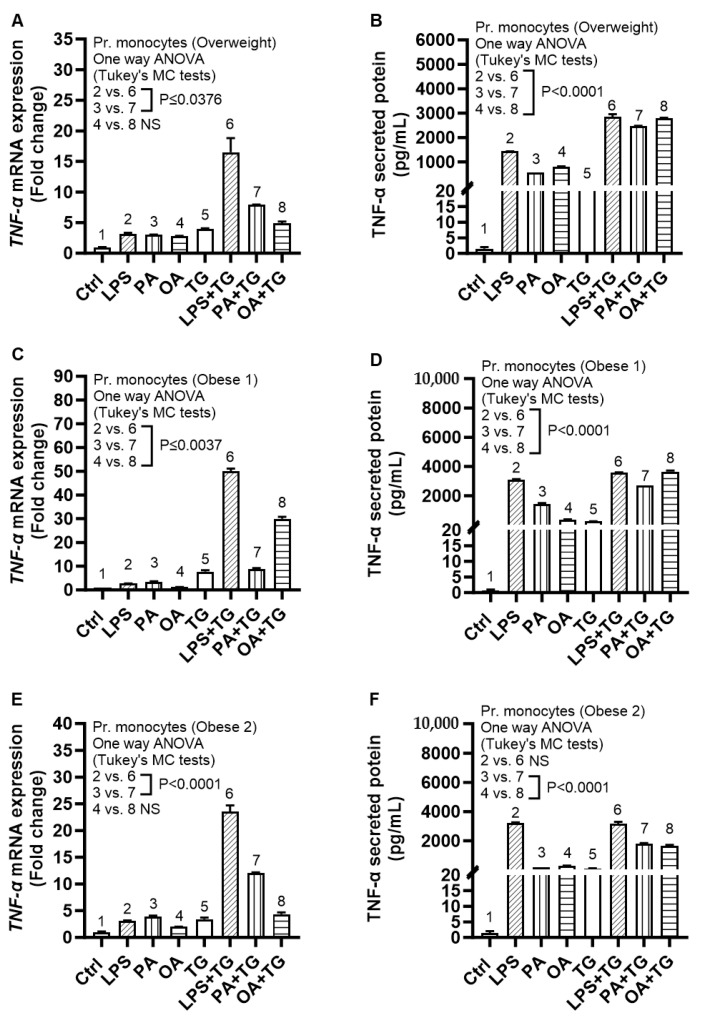
Increased TNF-α expression in primary monocytes isolated from overweight/obese individuals, following co-stimulation with metabolic (lipotoxic) and ER stresses. Primary monocytes were isolated from the peripheral blood mononuclear cells (PBMCs) of one overweight (BMI: 29.20 kg/m^2^) and two obese (BMI: 31.50 ± 0.1 kg/m^2^) individuals and monocytes were stimulated with LPS (10 ng/mL), PA (200 μM), and OA (200 μM), with or without ER stressor thapsigargin (TG) (1 μM), while cells treated with vehicle only represent control (Ctrl). TNF-α gene and secreted protein expression was determined using qRT-PCR and ELISA, respectively, as described in Section 4. Similar results were obtained from three independent experiments. Data (mean ± SEM) were analyzed using one-way ANOVA and group means were compared using Tukey’s multiple comparisons test. All *p*-values ≤ 0.05 were considered significant. The representative data show elevated TNF-α mRNA and secreted protein expression in: (**A**,**B**) overweight; (**C**,**D**) obese 1; and (**E**,**F**) obese 2 blood donors.

**Figure 9 ijms-24-15186-f009:**
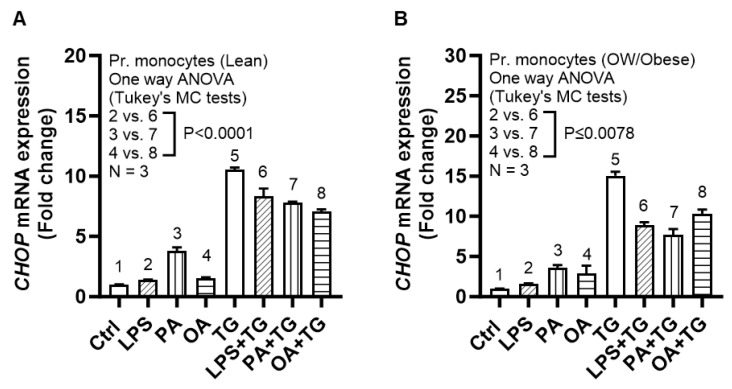
Metabolic/ER stresses upregulate *CHOP* mRNA expression in primary human monocytes from healthy lean and overweight/obese individuals. Primary monocytes were isolated from the peripheral blood samples from three healthy lean (BMI: 23.40 ± 0.35 kg/m^2^), one overweight (BMI: 29.20 kg/m^2^), and two obese (BMI: 31.50 ± 0.1 kg/m^2^) individuals. The cells plated in designated triplicate wells were stimulated with LPS (10 ng/mL), PA (200 μM), or OA (200 μM), in presence or absence of TG (1 μM) while control (Ctrl) wells were treated with vehicle (0.1% BSA) only. The cells were incubated for 24 h, total RNA was extracted and *CHOP* expression was determined using qRT-PCR as described in Section 4. Similar results were obtained from three independent experiments. Data (expressed as mean ± SEM) were analyzed using one-way ANOVA, Tukey’s multiple comparisons test, and *p*-values ≤ 0.05 were considered significant. The representative data show upregulated *CHOP* mRNA expression after co-stimulations with LPS+TG, PA+TG, or OA+TG, compared with respective controls without TG, in primary monocytes derived from blood of (**A**) lean and (**B**) overweight/obese individuals.

**Figure 10 ijms-24-15186-f010:**
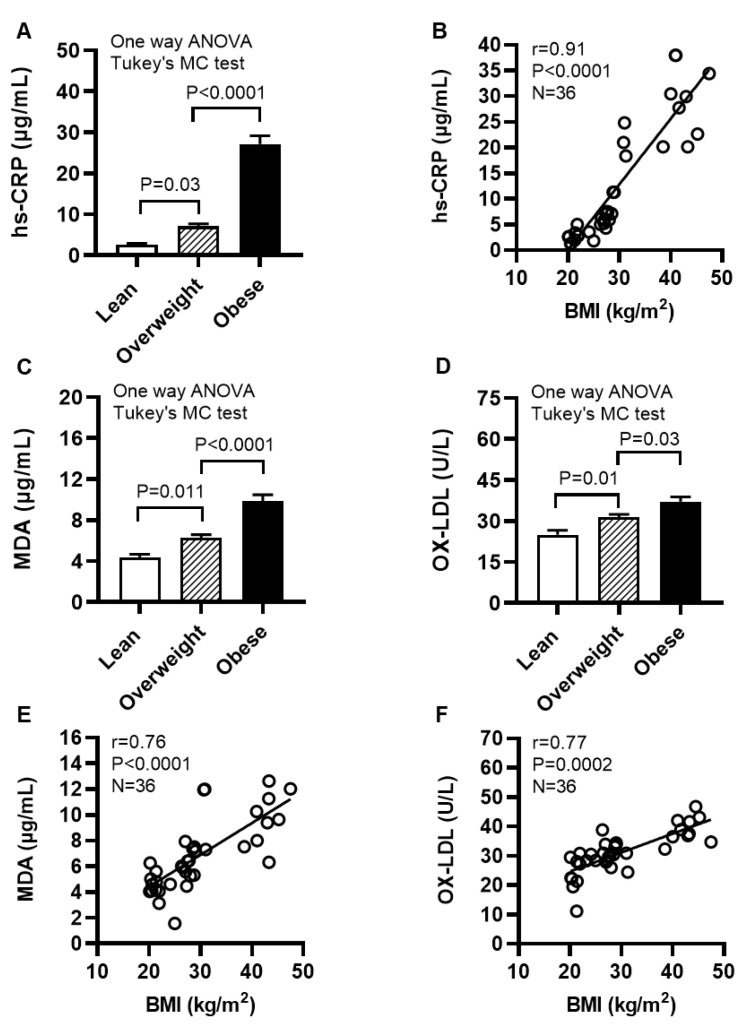
Individuals with obesity display increased expression of systemic inflammatory and oxidative stress biomarkers. Plasma samples were collected from lean (BMI: 21.69 ± 0.56 kg/m^2^), overweight (BMI: 27.71 ± 0.38 kg/m^2^), and obese (BMI: 39.51 ± 2.17 kg/m^2^) individuals (cohort 2), 12 each, and assessed the levels of high-sensitivity C-reactive protein (hs-CRP), malondialdehyde (MDA), and oxidized low-density lipoprotein (OX-LDL) using commercial kits, as described in Section 4. Similar results were obtained from three independent experiments. Data (expressed as mean ± SEM) were analyzed using one-way ANOVA and group means were compared using Tukey’s multiple comparisons test. Spearman correlation test was used to determine associations between variables. All *p*-values ≤ 0.05 were considered significant. (**A**) Increased hs-CRP levels are shown in obese, compared with lean and overweight counterparts (*p* < 0.0001). (**B**) hs-CRP levels were positively associated with BMI (r = 0.91, *p* ˂ 0.0001). (**C**) Elevated levels of MDA are shown in obese, compared with lean and overweight individuals (*p* < 0.0001). (**D**) Increased OX-LDL levels are shown in obese compared with lean and overweight counterparts (*p* ≤ 0.03). (**E**) MDA levels were positively correlated with BMI (r = 0.76, *p* ˂ 0.0001). (**F**) OX-LDL levels were positively associated with BMI (r = 0.77, *p* = 0.0002).

**Figure 11 ijms-24-15186-f011:**
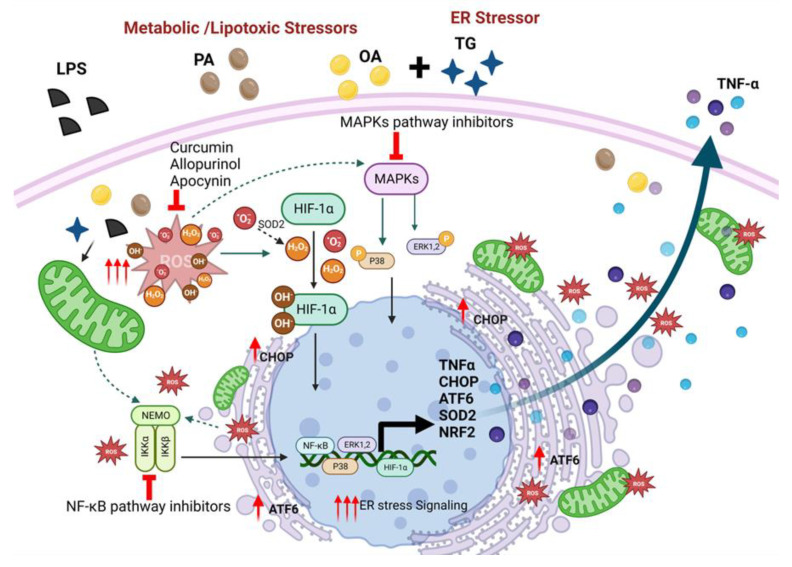
A schematic illustrating the cooperative mechanism between the ER and metabolic/lipotoxic stresses, leading to expression of TNF-α as well as markers representing ER stress (CHOP, ATF6) and antioxidant defense (SOD2, NRF2). ER: endoplasmic reticulum; LPS: lipopolysaccharide; PA: palmitate; OA: oleate; TG: thapsigargin; CHOP: C/EBP homologous protein; ATF6: activating transcription factor 6; SOD2: superoxide dismutase-2 (also known as MnSOD); NRF2: nuclear factor erythroid 2-related factor 2; HIF-1α: hypoxia-inducible factor 1-alpha; MAPKs: mitogen-activated protein kinases; NF-κB: nuclear factor kappa B. ROS: reactive oxygen species, e.g., superoxide (^•^O^−^_2_), hydrogen peroxide (H_2_O_2_), and hydroxyl radical (OH^−^). The figure was created using BioRender.com. Increased expression is shown by arrows (
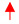
 or 

).

## Data Availability

All data are contained within the article and Appendix A.

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
