# Peer review of "Endoplasmic Reticulum Stress Promotes the Expression of TNF-α in THP-1 Cells by Mechanisms Involving ROS/CHOP/HIF-1α and MAPK/NF-κB Pathways"

_ijms, 2023, doi:10.3390/ijms242015186_

Round 1
Reviewer 1 Report
Reviewer comments and suggestions
The study investigated whether the endoplasmic reticulum (ER) stress in monocytic cells can contribute to amplifying metabolic inflammation; and the mechanism(s) involved. Metabolic stress was induced in THP-1 monocytic cells by treatments with lipopolysaccharide (LPS), palmitic acid (PA), or oleic acid (OA), in the presence or absence of ER stressor thapsigargin (TG). The study reported that a cooperative interaction between metabolic and ER stresses promoted TNF-α, ROS, CHOP, ATF6, SOD2, and NRF2 expression. Finally, they concluded that Individuals with obesity displayed increased adipose TNF-α gene/protein expression as well as elevated plasma levels of TNF-α, CRP, MDA, and OX-LDL (P≤0.05). Additionally, the interaction between metabolic and ER stresses, favors inflammation by triggering TNF-α production via the ROS/CHOP/HIF-1α and MAPK/NF-κB dependent mechanisms.
Overall, the manuscript was well written. However, a few concerns/comments needed to be explained/modified.
- Line 51-52 More references are needed to add here
- Line 72-73 Please check the reference that was cited in the MS-related to the information present
- Line 75 Please explain the metabolic in the introduction section to clear the pictures in the MS
- Line 76-77 Here the authors did not convince with the sentence why they hypothesis of this study, please explain it well
- Line 177 If they highlight metabolic stress,,, please explain the term in the introduction
- Line 192 I could not find any description of the antioxidant deference mechanism in the introduction.
- Line 437-438 I think it is known for monocytes cells.
- Comments for discussion section It would be nice if they present the flow chart of the mechanism involved along with the antioxidant used.
- Check the reference 1, 27,28,47,48,51
- Did the authors check the plagiarism report, kindly include the authenticate % for this manuscript.
Author Response
The authors thank the esteemed reviewer for a critical review and for comments/suggestions for improvement of the manusrcipt. Please see the rebuttal pdf attached here for your kind reference. We would be highly indebted for your kind approval and recommendation.

Reviewer 2 Report
In this study, Akhter et al. investigate how ER and metabolic stress increase TNF-α expression in THP-1 monocytic cells. They found that ER and metabolic stress lead to increased oxidative stress due to the production of ROS, which, in turn, increases the production of TNF-α. It was also determined by qPCR that this process involved several stress indicators, including CHOP, ATF6, SOD2, and NRF2. Conversely, treatment with antioxidants and ROS scavengers could reduce the production of TNF-α. Finally, the authors also found that higher TNF-α levels were observed in adipose tissue and blood of obsess individuals, as well as higher levels of inflammation and oxidative stress.
Overall, the study is comprehensive, with a lot of experimental data, and the text and the statistics are well-detailed. There are still several loopholes that the authors need to address with experiments for the final acceptance of the manuscript:
- First and foremost, the entire study is based on the THP-1 monocytic line. THP-1 being a leukemia cell line, stress-related studies can have innumerable variables. The study should be supplemented with primary cell line work. I understand that repeating experiments can be time-consuming. Hence the authors can perform only a couple of major experiments demonstrating ER stress leading to increased TNFa and the involvement of major pathways. In addition, the authors need to replace monocytic cells with THP-1 cells to prevent any misleading message.
- Figures 3 and 4 show gene expression changes in stress pathway markers. These data should be supplemented with protein expression as well. The major reason for this is that there are a lot of changes during the translational and post-translational levels in stress conditions. Figure 3 should also include PERK qPCR and western blot for PERK, phospho-PERK, IRE-1a, phospho-IRE-1a, ATF6, XBP1s
- Figures 7 and 8 are randomly added without a significant rationale. Furthermore, figure 8 does not add any value to the manuscript since it has already been well-established in the literature. Instead, the authors should collect blood from healthy vs. obese individuals, isolate monocytes, and test ER stress and TNFa levels in those cells.
Author Response
The authors thank the esteemed reviewer for a critical review and for comments/suggestions for improvement of the manuscript. Please see the rebuttal pdf attached here for your kind reference. We would be highly indebted for your kind approval and recommendation.

Round 2
Reviewer 1 Report
No more comments
Author Response
Review Report Form 1
Comments and Suggestions for Authors
No more comments
The authors are highly grateful to the esteemed reviewer for taking time to review and kindly approve the revised manuscript.
Submission Date
18 July 2023
Date of this review
09 Oct 2023 19:12:27

Reviewer 2 Report
The authors have done a commendable job in addressing all the concerns. Including protein data as well primary cell data certainly raises the bar of the manuscript and significantly enhances the impact of the findings. After observing the new data, there is one minor concern regarding these statements "Additionally, IRE1α protein phosphorylation was also determined and no significant changes were detected between treatments (Supplementary Fig S2). Taken together, PA+TG stimulation upregulates the multiple pathways of ER stress in THP-1 cells."
These two statements seem contradicting. The authors could address the concern by taking out the 'Supplementary Fig2' statement and insert it in their limitations section. Here they could explain their efforts on exploring ER stress pathways at the protein levels, the difficulties they encountered and how in the future studies it will be critical it is to delve deeper into the translational and posttranslational studies of ER stress pathways. This modification would keep the message clearer for 'figure 3' albeit at the transcriptional level.
Author Response
The authors are greatly obliged and thankful to the esteemed reviewer for kindly accepting and recommending the revised manuscript. In addressing the concerns above, the statement related to supplementary Fig S2 has been removed from the text (please see lines 190-192) and, accordingly, Fig S2 is removed from the supplementary material file. As kindly advised, we have now added lack of the translational data regarding ER stress markers as a caveat in the study; an aspect to be addressed in future investigations (please see lines 686-693). Hope it will meet your kind approval.
Submission Date
18 July 2023
Date of this review
09 Oct 2023 14:59:50